# tRNA ligase structure reveals kinetic competition between non-conventional mRNA splicing and mRNA decay

**Jirka Peschek\*, Peter Walter**

Department of Biochemistry and Biophysics, Howard Hughes Medical Institute, University of California, San Francisco, San Francisco, United States

**Abstract** Yeast tRNA ligase (Trl1) is an essential trifunctional enzyme that catalyzes exon-exon ligation during tRNA biogenesis and the non-conventional splicing of *HAC1* mRNA during the unfolded protein response (UPR). The UPR regulates the protein folding capacity of the endoplasmic reticulum (ER). ER stress activates Ire1, an ER-resident kinase/RNase, which excises an intron from *HAC1* mRNA followed by exon-exon ligation by Trl1. The spliced product encodes for a potent transcription factor that drives the UPR. Here we report the crystal structure of Trl1 RNA ligase domain from *Chaetomium thermophilum* at 1.9 Å resolution. Structure-based mutational analyses uncovered kinetic competition between RNA ligation and degradation during *HAC1* mRNA splicing. Incompletely processed *HAC1* mRNA is degraded by Xrn1 and the Ski/exosome complex. We establish cleaved *HAC1* mRNA as endogenous substrate for ribosome-associated quality control. We conclude that mRNA decay and surveillance mechanisms collaborate in achieving fidelity of non-conventional mRNA splicing during the UPR.
DOI: https://doi.org/10.7554/eLife.44199.001

## Introduction

RNA ligases are found in all domains of life. They catalyze the ligation of RNA molecules via phosphodiester bonds during different RNA processing events, such as repair, editing and splicing (*Popow et al., 2012*). The fungal tRNA ligase Trl1 (previously named Rlg1) is encoded by an essential gene and is involved in tRNA splicing and the unfolded protein response (UPR). Trl1 is a tripartite enzyme (*Figure 1A*), consisting of an N-terminal adenylyltransferase domain (ligase; LIG), which belongs to the nucleotidyltransferase superfamily alongside DNA ligases and RNA capping enzymes, a C-terminal cyclic phosphodiesterase domain (CPD) and a central polynucleotide kinase (KIN) domain (*Phizicky et al., 1986*; *Xu et al., 1990*). Trl1 substrates initially contain a 2′,3′ cyclic phosphate and a 5′-OH group at the RNA termini (*Greer et al., 1983*). The ligation reaction progresses via three enzymatic steps (*Figure 1B*). First, the CPD activity opens the 2′,3′ cyclic phosphate by hydrolysis to form a 3′-OH/2′ phosphate terminus, and, second, the KIN activity phosphorylates the 5′-OH in an NTP-dependent reaction – preferring GTP over ATP. These first two steps 'heal' (i.e., modify) the RNA termini in preparation for the ligation reaction. Third, the LIG activity 'seals' the healed ends through ATP-dependent phosphodiester bond formation. This final reaction occurs in three nucleotidyl transfer steps: (1) Trl1-LIG reacts with ATP to form a covalent LIG-(lysyl-N)-AMP intermediate; (2) the bound AMP is transferred to the 5′ phosphate end to form a 5′-to-5′ RNA-adenylate; (3) Trl1-LIG catalyzes the attack by the 3′-OH on the RNA-adenylate to form a phosphodiester bond, releasing AMP (*Greer et al., 1983*; *Phizicky et al., 1986*). The remaining 2′ phosphate at the splice junction is removed by an additional enzyme, Tpt1, which is a 2′ phosphotransferase that, like Trl1, is also essential for cell viability (*Banerjee et al., 2019*; *Culver et al., 1997*; *Culver et al., 1993*).

\*For correspondence:
jirka@walterlab.ucsf.edu

**Competing interests:** The authors declare that no competing interests exist.

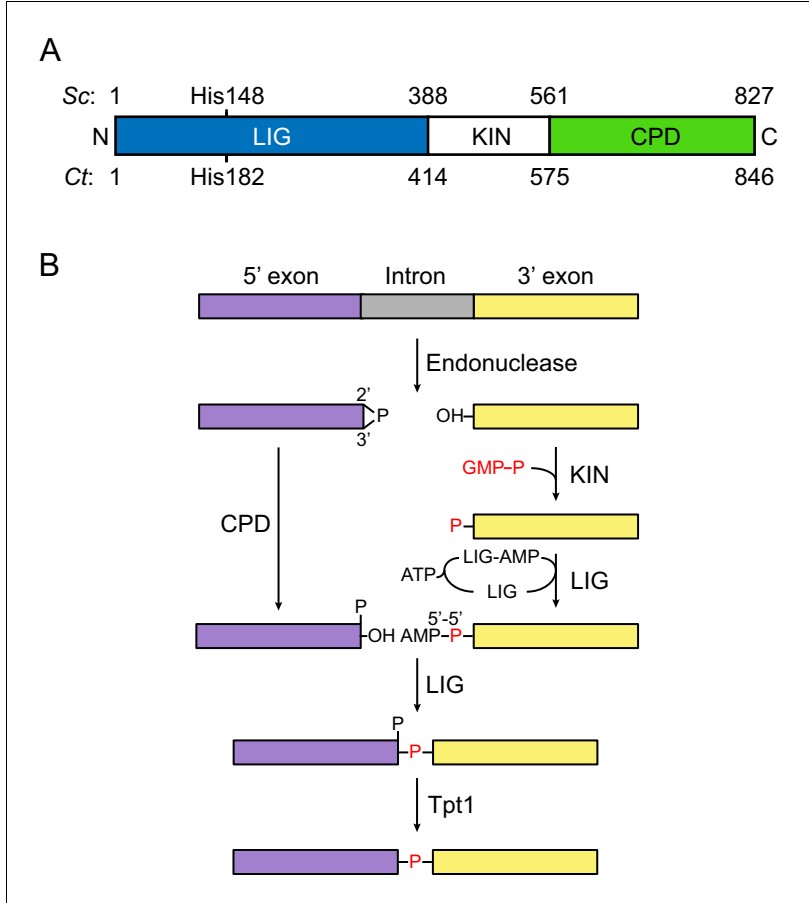

**Figure 1.** Fungal tRNA ligase Trl1. (**A**) Domain organization of Trl1. The ligase/adenylyltransferase domain (LIG) is shown in blue, the polynucleotide kinase domain (KIN) in white, the cyclic phosphodiesterase domain (CPD) in green. The residue numbering refers to the domain boundaries of Trl1 from *Saccharomyces cerevisiae* (*Sc*) and *Chaetomium thermophilum* (*Ct*). The relative position of the mutated histidine in the UPR mutant is indicated. See *Figure 1—figure supplement 1* for a sequence alignment of Trl1-LIG from both species. (**B**) Trl1-mediated non-conventional RNA splicing mechanism. The RNA is cleaved at the exon-intron junctions by an endonuclease (SEN or Ire1), which leaves a 2',3'-cyclic phosphate end on the 5' exon (purple) and a 5'-OH on the 3' exon (yellow). After removal of the intron (gray), the exon ends are first modified ('healed') by the CPD and KIN domains and finally ligated ('sealed') by the LIG domain. Exon ligation progresses via a covalent ligase-AMP intermediate (LIG-AMP) and a 5'−5' RNA-adenylate. The residual 2'-phosphate at the splice site is then removed by a separate enzyme, the 2'-phosphotransferase Tpt1.

DOI: https://doi.org/10.7554/eLife.44199.002
The following figure supplement is available for figure 1:

**Figure supplement 1.** Sequence alignment between *sc*Trl1 and *ct*Trl1.
DOI: https://doi.org/10.7554/eLife.44199.003

Trl1 ligates tRNA halves after intron excision by the tRNA splicing endonuclease (SEN) complex (*Greer et al., 1983*; *Peebles et al., 1983*). In addition, it is a key component of the UPR, a major intracellular stress signaling pathway (*Sidrauski et al., 1996*). All eukaryotic cells monitor and adjust the protein folding capacity of their endoplasmic reticulum (ER) through the UPR signaling network. The evolutionarily most conserved – and in fungi sole – branch of the UPR signals via a unique, non-conventional mRNA splicing reaction: Protein folding stress in the ER activates the cytosolic endonuclease domain of the transmembrane stress sensor Ire1 (*Cox et al., 1993*; *Mori et al., 1993*). The activated RNase cleaves *HAC1* mRNA in fungi or *XBP1* mRNA in metazoans at the splice sites (*Calfon et al., 2002*; *Cox and Walter, 1996*; *Yoshida et al., 2001*). Next, a conformational change within the RNA actively ejects the intron and coordinates the two exons (*Gonzalez et al., 1999*;

*Peschek et al., 2015*), which are ligated by Trl1 in fungi (*Sidrauski et al., 1996*) or the RTCB ligase complex in metazoans (*Jurkin et al., 2014*; *Kosmaczewski et al., 2014*; *Lu et al., 2014*). The spliced *HAC1* and *XBP1* mRNAs are then translated to produce the active transcription factors Hac1 and XBP1 that drive expression of UPR target genes in yeast and metazoan cells, respectively (*Cox and Walter, 1996*; *Mori et al., 1996*; *Nikawa et al., 1996*).

Here, we report the crystal structure of the Trl1 ligase domain from *Chaetomium thermophilum*. Guided by our structural data, we provide evidence that the non-conventional splicing of *HAC1* mRNA competes with RNA decay in the cell. We further establish cleaved *HAC1* mRNA as an endogenous substrate for ribosome-associated mRNA quality control.

## Results

### *Chaetomium thermophilum* Trl1

Earlier studies identified several residues within yeast tRNA ligase Trl1 that are critical for its essential function in tRNA splicing. Intriguingly, a yeast genetic screen in *Saccharomyces cerevisiae* (*sc*) revealed a single His(148)Tyr point mutation lying within Trl1-LIG that in vivo abolished the UPR signaling yet did not impair tRNA splicing. This mutant allele (henceforth referred to as *trl1-H148Y*, originally named *rlg1-100*) paved the way to discovery of Trl1 as the RNA ligase during Ire1-mediated splicing of *HAC1* mRNA (*Sidrauski et al., 1996*). To understand the structural basis for the two distinct roles of Trl1, we sought to crystallize the full-length enzyme, as well as functional modules of it. While our efforts failed using *S. cerevisiae*-derived protein, we were successful using protein derived from *Chaetomium thermophilum* (*ct*), a thermophilic fungus that previously proved invaluable for structural studies (*Amlacher et al., 2011*; *Bock et al., 2014*). *ct*Trl1 has the same tripartite domain structure as *sc*Trl1 (*Figure 1A*) with 39% overall sequence identity (*Figure 1—figure supplement 1*). Moreover, expression of *ct*Trl1 rescued *S. cerevisiae trl1Δ* cells from the lethal effects of the deletion, both under normal growth conditions as well as during ER stress induced by tunicamycin (Tm), an inhibitor of N-linked glycosylation (*Figure 2A*). Hence, *ct*Trl1 catalyzes in *S. cerevisiae* cells the ligation of both tRNA halves, thus sustaining normal growth, and *HAC1* mRNA exons during the UPR. In addition, recombinantly expressed *ct*Trl1 efficiently ligated the exons of a *HAC1*-derived RNA substrate, $HAC1^U$-508 ('U' for unspliced, that is intron containing), after its endonucleolytic cleavage by Ire1 completing the splicing reaction in vitro (*Figure 2B*).

### Structure of the *ct*Trl1 ligase domain

Aiming toward structural analysis, we expressed *ct*Trl1 constructs of varying length with incorporated selenomethionine in *Escherichia coli* and purified the proteins. While the full-length protein resisted crystallization, we obtained diffracting crystals belonging to space group $P2_12_12_1$ of *ct*Trl1-LIG (residues 1 to 414) bound to the α,β-non-hydrolyzable ATP analog α,β-methyleneadenosine 5′-triphosphate (AMPcPP). The structure was solved by SeMet-SAD phasing to 1.9 Å resolution (see *Table 1* for data collection and refinement statistics). The overall structure delineates two subdomains: an N-terminal (LIG-N; aa 13–326) and a C-terminal (LIG-C; aa 327–407) subdomain (*Figure 3A*, colored blue and yellow, respectively). LIG-N is composed of three central antiparallel β-sheets surrounded by seven α-helices. The overall architecture of this domain resembles the bacteriophage T4 RNA ligase (T4Rnl1) structure consistent with it carrying the adenylyltransferase activity (*El Omari et al., 2006*). The most N-terminal residues (~35 aa), which wrap around the domain, are absent in T4Rnl1. LIG-C has an all-helical fold that is unrelated, based on structural comparison by Dali (*Holm and Laakso, 2016*), to all other structures in the PDB from the same superfamily, which besides ATP-dependent RNA ligases also includes DNA ligases and mRNA capping enzymes.

LIG's active site is buried in the center of a positively charged cleft that spans LIG-N and is extended by LIG-C (*Figure 3B*). The adenosine nucleotide binding pocket is contained in LIG-N. In our structure, Nζ of Lys148 in the highly conserved nucleotide binding motif I [Kx(D/H/N)G] (*Figure 1—figure supplement 1*), which is the site of covalent AMP attachment, is more than 4 Å away from the α-phosphonate (*Figure 3C*), suggestive of conformational rearrangements necessary for the formation of the Trl1-(lysyl-N)-AMP intermediate. Further analysis of the nucleotide binding site identified the residues interacting with AMPcPP. Thr146, Leu147, Glu149, Arg305 and Lys325 form contacts with the adenine base via hydrogen bonding, and His241 via π-stacking. The adenosine

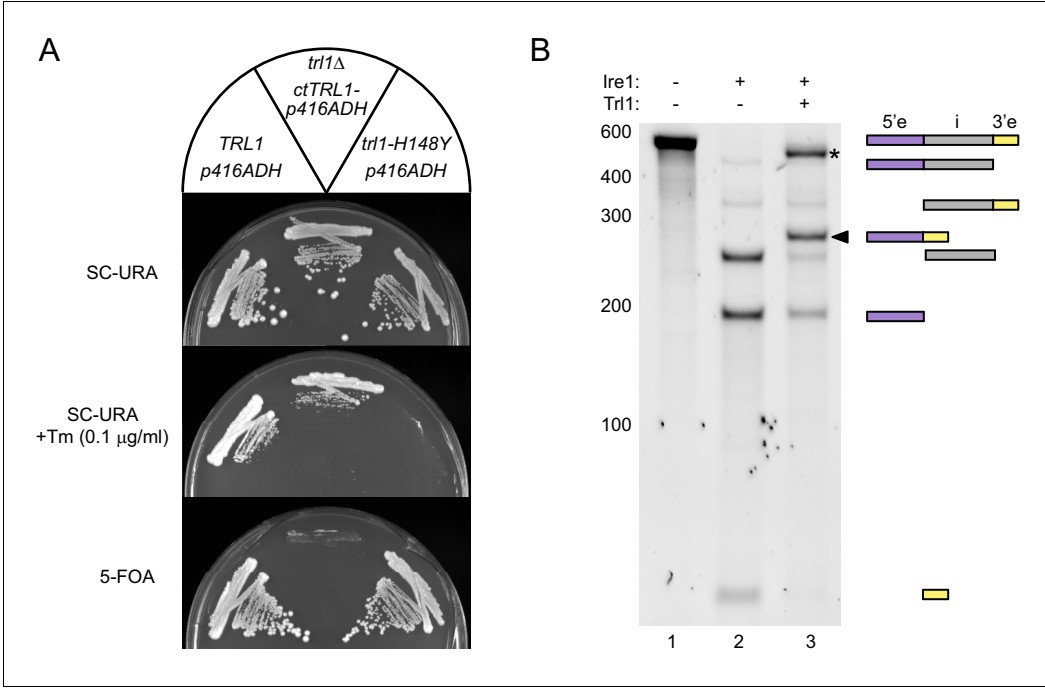

**Figure 2.** Functional complementation by thermophilic *ct*Trl1. (**A**) Functional complementation in *S. cerevisiae trl1Δ* by *ct*Trl1. Expression of *ct*Trl1 from a plasmid (*URA3* selection) permitted growth of a *trl1Δ* strain under normal growth conditions (SC-URA) and ER stress (SC-URA +Tm). Counterselection against the *URA3*-containing plasmid using 5'-fluoroorotic acid (5-FOA) confirmed the essential function of Trl1. The respective *TRL1* genotype and plasmid are indicated in the key on top. (**B**) In vitro splicing of a *HAC1*-derrived RNA substrate was analyzed by denaturing urea-PAGE. The input RNA (lane 1) was first cleaved by *sc*Ire1 yielding three dominant bands for the separated exons and the intron (lane 2). Faint bands corresponding to single cleaved RNA were observed. Addition of *ct*Trl1 completed the splicing reaction resulting in covalently ligated exons (lane 3, marked by a closed arrowhead). An additional ligation product is marked with an asterisk. RNase H digestion identified the band as circularized tandem *HAC1* intron RNA (see **Figure 2—figure supplement 1**). The position on the gel and relative length of the RNA species is depicted by icons on the right using the same color code as in **Figure 1B**.

DOI: https://doi.org/10.7554/eLife.44199.004

The following figure supplement is available for figure 2:

**Figure supplement 1.** RNase H assay.

DOI: https://doi.org/10.7554/eLife.44199.005

nucleoside base of AMPcPP is in *syn* conformation. Arg99 functions as hydrogen bond donor with the 3' oxygen of the ribose ring and together with Lys169 forms salt-bridges to the γ-phosphate (**Figure 3C**).

Two conserved residues, Arg334 and Arg337, within an α-helix (α11) of the LIG-C subdomain coordinate a single well-ordered sulfate ion via salt bridges near the active site pocket. In addition, His227 is within distance to form a third salt bridge, and Asn150 forms a hydrogen bond with the sulfate ion (**Figure 3D**). Notably, Arg337 and His227 have been previously identified as essential residues by alanine scanning mutagenesis (**Wang and Shuman, 2005**). We surmise that the sulfate ion represents a surrogate for one of the phosphate groups of Trl1's RNA substrate.

## The UPR-disruptive *trl1-H148Y* mutant possesses reduced ligation activity

Based on the strong homology of *S. cerevisiae* and *C. thermophilum* Trl1 (**Figure 1—figure supplement 1**), we found all residues previously identified to be essential for the RNA ligase activity of *sc*Trl1 conserved between both species. When we mapped these residues on the structure of *ct*Trl1-LIG (**Figure 4A**), we found that they all cluster around the active site. In addition, these residues are all surface exposed, which points to their importance to binding and correct positioning of RNA

**Table 1.** Data collection and refinement statistics.

| Datasets: | High-resolution (remote) | Se SAD phasing (peak) |
|---|---|---|
| **Data collection** | | |
| Beamline | APS 23ID-D | APS 23ID-D |
| Wavelength (Å) | 1.0332 | 0.9793 |
| Space group | $P2_12_12_1$ | $P2_12_12_1$ |
| Cell dimensions a, b, c (Å) $\alpha, \beta, \gamma$ (°) | 49.73, 56.58, 173.88 90, 90, 90 | 49.63, 56.46, 173.87 90, 90, 90 |
| Resolution (Å) | 37.74–1.90 (1.94–1.90) | 49.63–2.50 (2.60–2.50) |
| Total reflections | 159936 (10248) | 175374 (20071) |
| Unique reflections | 39302 (2495) | 17561 (1930) |
| Multiplicity | 4.1 (4.1) | 10.0 (10.4) |
| Completeness (%) | 99.3 (99.1) | 99.1 (98.1) |
| $I/\sigma(I)$ | 10.3 (1.2) | 21.9 (7.3) |
| CC(1/2) | 0.998 (0.793) | 0.999 (0.982) |
| $R_{pim}$ | 0.036 (0.575) | 0.024 (0.090) |
| **Phasing** | | |
| Number of Se sites | | 2 |
| Figure of merit | | 0.64 |
| **Refinement** | | |
| $R_{work}/R_{free}$ | 0.170/0.214 | |
| Number of atoms | 3457 | |
| Wilson B factor (Å) | 33.8 | |
| RMS deviations Bonds (Å) Angles (°) | 0.007 1.386 | |
| Ramachandran plot % favored % allowed % outliers | 97.91 1.83 0 | |
| PDB ID | 6N67 | |

Values in parentheses refer to the highest resolution shell.

The $R_{free}$ set consists of 5% randomly chosen data excluded from refinement.

DOI: https://doi.org/10.7554/eLife.44199.007

substrates. Interestingly, His148 was also identified as an essential residue as its change to Ala, Asn, or Gln scored as lethal in vivo (*Wang and Shuman, 2005*). By contrast, cells bearing the *sc-trl1-H148Y* allele were viable and shown previously to block UPR-induced nonconventional mRNA splicing (*Sidrauski et al., 1996*). This conserved histidine (His182 in *ct*Trl1) is surface-exposed and located on the periphery of the active-site groove in the vicinity of other essential residues (*Figure 4A*, colored red). Since the positioning of His182 suggests its participation in substrate binding, we performed in vitro ligation assays with recombinant, purified wild-type (WT) and *sc*Trl1-LIG-H148Y. As substrate we used Ire1-cleaved *HAC1*[U]-508 RNA whose ends were enzymatically 'healed' (modified) to yield an appropriate substrate for Trl1-LIG (see Materials and methods for details). Equivalent concentrations of the WT enzyme yielded more spliced (i.e. ligated) product of the *HAC1*[U]-508 RNA substrate compared to *sc*Trl1-LIG-H148Y (*Figure 4B*). These results indicate that *sc*Trl1-H148Y is catalytically impaired towards a *HAC1*-derived RNA substrate but not inactive. To test if reduced substrate affinities cause the inhibiting effect of the H148Y mutation on RNA ligation, we determined the dissociation constants ($K_D$) of *sc*Trl1-LIG-WT and *sc*Trl1-LIG-H148Y towards *HAC1*-derived 5' and 3' exon RNA oligonucleotides using fluorescence titration experiments. Despite the reduced ligation efficiency of the H148Y mutant, we only observed a small impact on binding affinity towards the RNA substrates (*Figure 4—figure supplement 1*). To obtain a

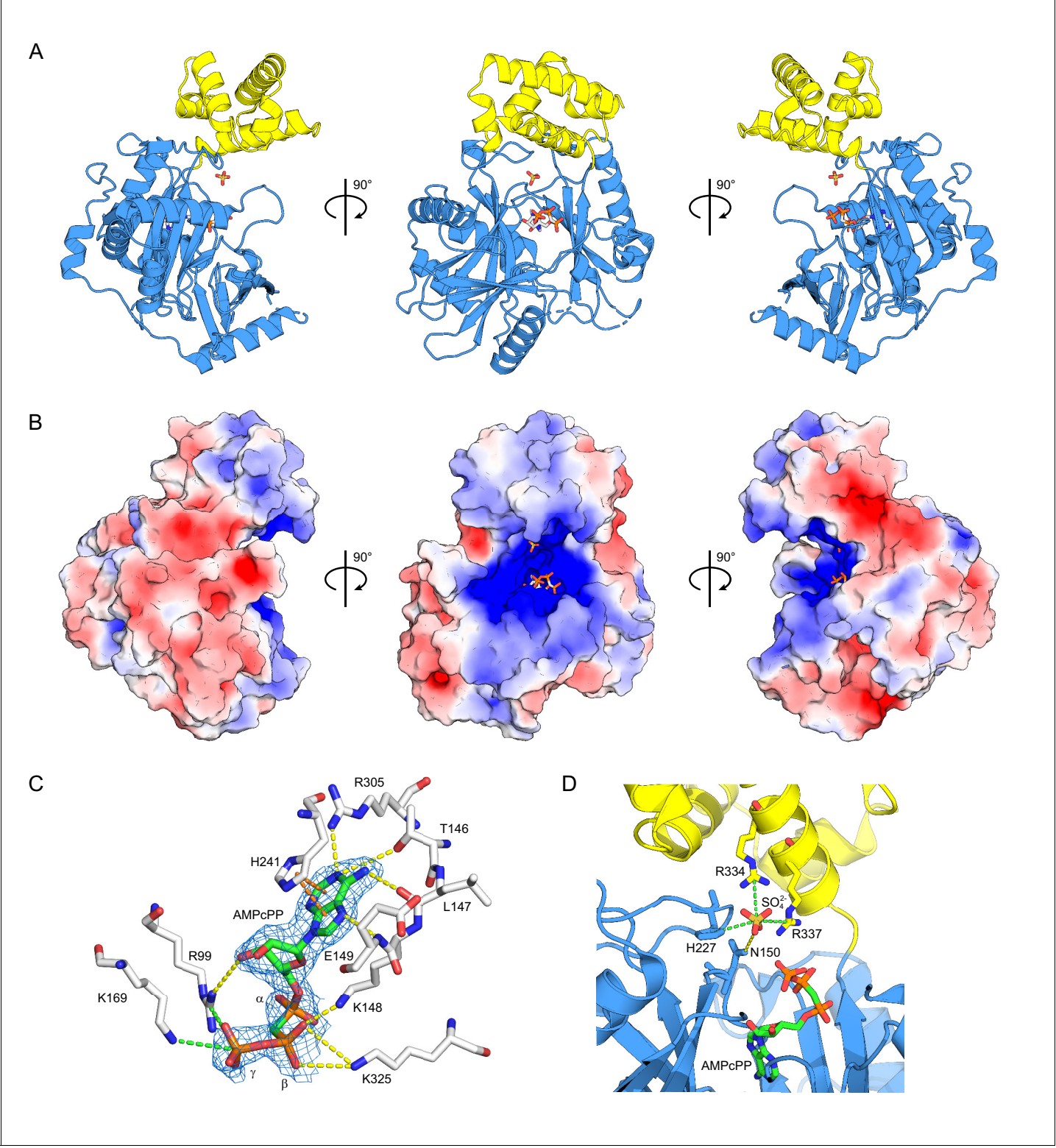

**Figure 3.** Structure of *ct*Trl1-LIG. (**A**) Cartoon representation of the overall structure of *ct*Trl1-LIG from three views. The adenylyltransferase domain is colored in blue and the C-terminal domain in yellow. AMPcPP and the sulfate ion near the active site are depicted as sticks. (**B**) Electrostatic surface potential map of *ct*Trl1-LIG. Positive potential is represented in blue and negative potential in red. (**C**) ATP binding pocket with bound AMPcPP. The interacting amino acid residues and AMPcPP are shown as sticks. Atomic contacts between *ct*Trl1-LIG and AMPcPP are indicated by dashed lines, with yellow for hydrogen bonds, orange for π-stacking and green for salt bridges. The electron density around AMPcPP (blue mesh) is extracted from a

*Figure 3 continued on next page*

*Figure 3 continued*

composite omit map ($2mF_o\text{-}DF_c$), contoured at 1 σ. (**D**) Sulfate ion binding near the active site. Amino acids, AMPcPP and the sulfate ion are shown as sticks. C atoms are colored according to the respective domains (see **A**) in blue (adenylyltransferase domain) or yellow (C terminal domain). Atomic contacts are depicted as in **C**.

DOI: https://doi.org/10.7554/eLife.44199.006

quantitative comparison of the differences in ligation kinetics, we tested both enzymes in an RNA oligonucleotide ligation assay. To this end, we monitored ligation of a fluorescein (FAM)-labeled *HAC1*-derived 5′ exon RNA oligonucleotide (last 10 nucleotides before the splice site; enzymatically modified 2′ phosphate/3′ OH end) to an unlabeled 3′ exon RNA oligonucleotide (first 20 nucleotides after the splice site; harboring a 5′ phosphate). The isolated ligase domain (*sc*Trl1-LIG-WT or -H148Y) was pre-incubated with the 3′ exon fragment to improve the reaction efficiency. The pre-formed LIG/3′ exon complex was used in excess over the FAM-labeled 5′ exon fragment and the time-course of ligation was analyzed by denaturing urea-PAGE (*Figure 4C*). The results were fitted to a first-order model (*Figure 4D*) and revealed a 2.2-fold longer half-life for the H148Y mutant ($t_{1/2}$ = 9.2 min) compared to WT ligase ($t_{1/2}$ = 4.1 min). These kinetic data confirmed the impeding effect of the His-to-Tyr mutation on *HAC1* exon-exon ligation. Interestingly, the mutant ligase did not only display slower ligation kinetics but also plateaued at a lower amount of ligation product (8%) compared to wild-type enzyme (50%).

## Trl1 competes with RNA decay pathways during *HAC1* mRNA splicing

To explain how the reduced ligase activity of Trl1-H148Y would impair *HAC1* mRNA splicing so severely in cells as to entirely block UPR signaling, we surmised that, in the presence of RNA decay machinery in vivo, the balance between exon ligation and RNA decay might be tipped towards decay. To test this notion directly, we expressed Trl1-H148Y from the strong constitutive alcohol dehydrogenase 1 (*ADH1*) promoter. Both *sc*Trl1-H148Y and *ct*Trl1-H182Y allowed growth on Tm-containing medium, as did expression of the corresponding LIG only construct (*Figure 5A*, compare sector 1 to 2 and 4, and sector 1 to 3 and 5, respectively). The suppression of the mutant phenotype emphasizes that the His148-to-Tyr mutation does not fully inhibit Trl1-LIG functionality and that it is indeed the reduced enzymatic activity of the LIG domain only that becomes limiting, causing the UPR-deficient phenotype of *trl1-H148Y* cells.

Second, we tested whether unligated, Ire1-cleaved fragments of the *HAC1* mRNA are substrates for RNA decay pathways using yeast growth assays (*Figure 5B*). While the capped 5′ exon represents a possible substrate for 3′-to-5′ exonucleases, the 3′ exon, which still has a poly(A) tail, hence, would likely be degraded in the 5′-to-3′ direction (*Figure 5C*). The two major conserved exonucleolytic RNA decay enzymes are Xrn1 (5′-to-3′) and the multi-protein RNA exosome complex (3′-to-5′). Since deletion of both pathways is synthetically lethal in yeast, we tested their contribution individually by deletion of *XRN1* and *SKI2* (encoding for an RNA helicase in the exosome-assisting Ski complex). Indeed, individual deletion of either *XRN1* or *SKI2* rescued growth of *trl1-H148Y* cells upon ER stress (*Figure 5B*). While growth of *ski2Δ* cells was indistinguishable from that of WT cells, *xrn1Δ* cells exhibited a reduced colony size, which might be explained by the annotated slow growth phenotype of *XRN1* deletion being exacerbated in the presence of ER stress. To ensure that the observed rescue indeed resulted from restored UPR signaling, we generated *HAC1* deletions in the respective strains and tested their growth. Neither the *xrn1Δ hac1Δ* nor the *ski2Δ hac1Δ* strain were viable in the presence of ER stress (*Figure 5—figure supplement 1*). The rescue by deletion of the major cytosolic RNA degradation pathway in either the 5′-to-3′ or the 3′-to-5′ direction substantiates that non-conventional splicing and RNA decay are competing outcomes after Ire1 cleavage of *HAC1* mRNA.

## Deletion of *XRN1* and *SKI2* rescues *trl1-H148Y* mutant cells by different mechanisms

As mentioned above, the cleaved *HAC1* exons should only be susceptible from one end to the attack by exonucleases (*Figure 5C*); yet, deletion of RNA decay in only one direction was sufficient to allow a *HAC1*-dependent rescue of the *trl-H148Y* phenotype. To understand the mechanism of

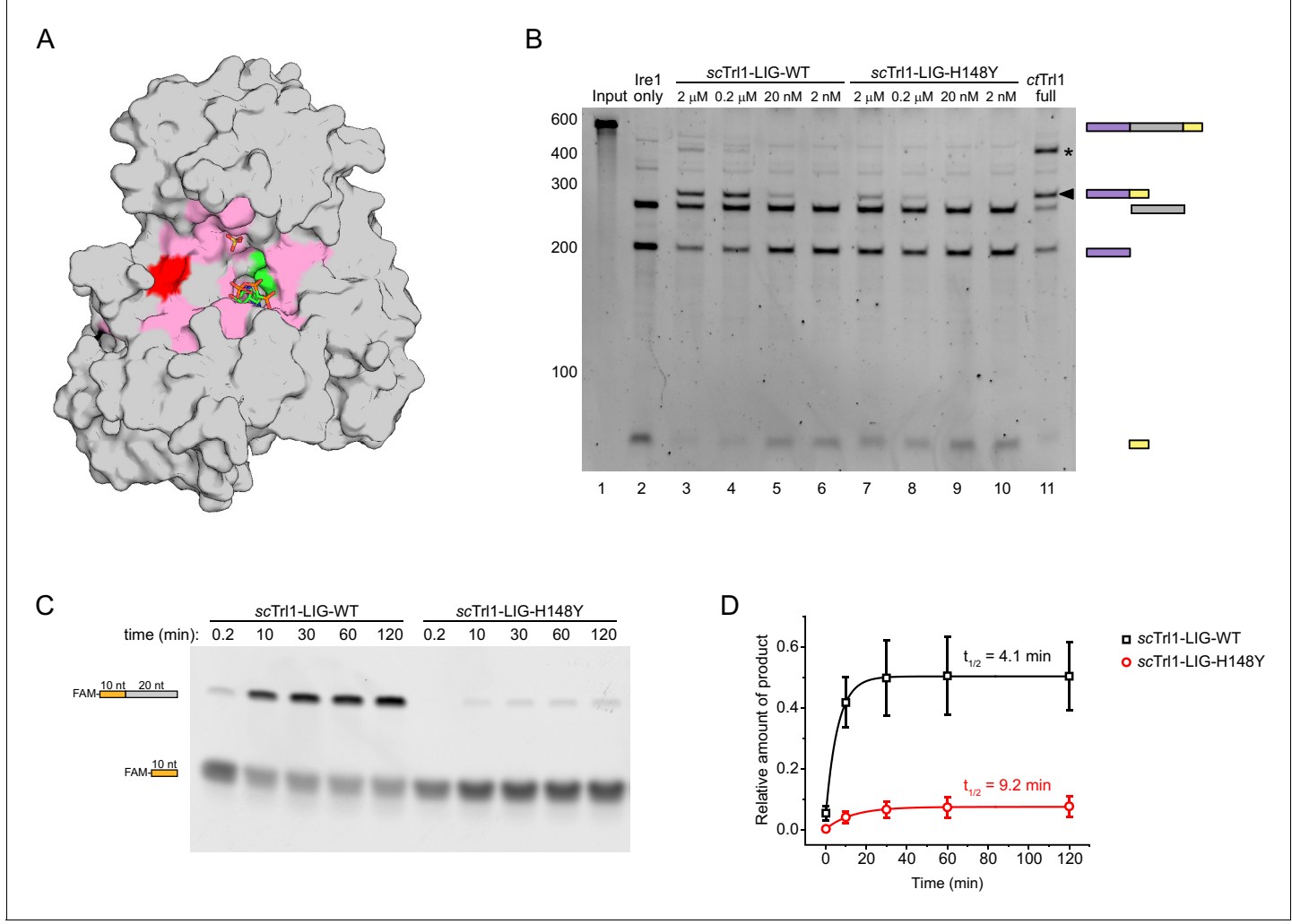

**Figure 4.** The UPR mutant *trl1-H148Y* compromises RNA ligation kinetics of Trl1. (A) Functionally important residues of Trl1-LIG. The structure of *ct*Trl1-LIG is depicted as surface model, AMPcPP and the sulfate ion are shown as sticks. The active site Lys is depicted in green, His182 (equivalent to His148 in *S. cerevisiae*) in red and residues that have been previously identified as essential by mutagenesis in pink (*Wang and Shuman, 2005*). (B) In vitro splicing of a *HAC1*-derrived RNA substrate was analyzed by denaturing urea-PAGE. The input RNA (lane 1) was first cleaved by *sc*Ire1 yielding three dominant bands for the separated exons and the intron (lane 2). Ligation efficiency was compared by titrating purified *sc*Trl1-LIG-WT (lanes 3–6) or *sc*Trl1-LIG-H148Y (lanes 7–10) to the splicing reaction. Full-length *ct*Trl1 was used as positive control (lane 11). An additional ligation product is marked with an asterisk. RNase H digestion identified the band as circularized *HAC1* intron RNA (see *Figure 2—figure supplement 1*). (C) Time-course denaturing urea-PAGE of RNA oligonucleotide ligation by WT or H148Y mutant *sc*Trl1-LIG. The 5' fluorescein (FAM)-labeled 10 nt *HAC1* 5' exon oligonucleotide (depicted in orange, lower band) and its ligation product (upper band) with a 20 nt *HAC1* 3' exon oligonucleotide (depicted in gray) was monitored using fluorescence detection. A representative gel out of three replicates is shown. (D) Quantification of the RNA ligation assay in C. The relative amounts of ligated product in the presence of *sc*Trl1-LIG-WT (black squares) or *sc*Trl1-LIG-H148Y (red circles) were fitted to a first-order model to determine the depicted half-lives. Values represent mean and standard deviation of technical replicates (n = 3).

DOI: https://doi.org/10.7554/eLife.44199.008

The following figure supplement is available for figure 4:

**Figure supplement 1.** RNA binding assay.

DOI: https://doi.org/10.7554/eLife.44199.009

rescue, we used reverse transcription (RT-)PCR to monitor *HAC1* mRNA splicing in these cells. To this end, we used oligonucleotide primers in both exons flanking the intron to obtain differently sized amplification products for unspliced and spliced *HAC1* mRNA based on the presence or absence of the 252 nt intron. As expected, *HAC1* mRNA was spliced upon Tm-induced ER stress in the WT strain and not affected by deletion of *XRN1* or *SKI2* (*Figure 5D*, lanes 1–6). In agreement

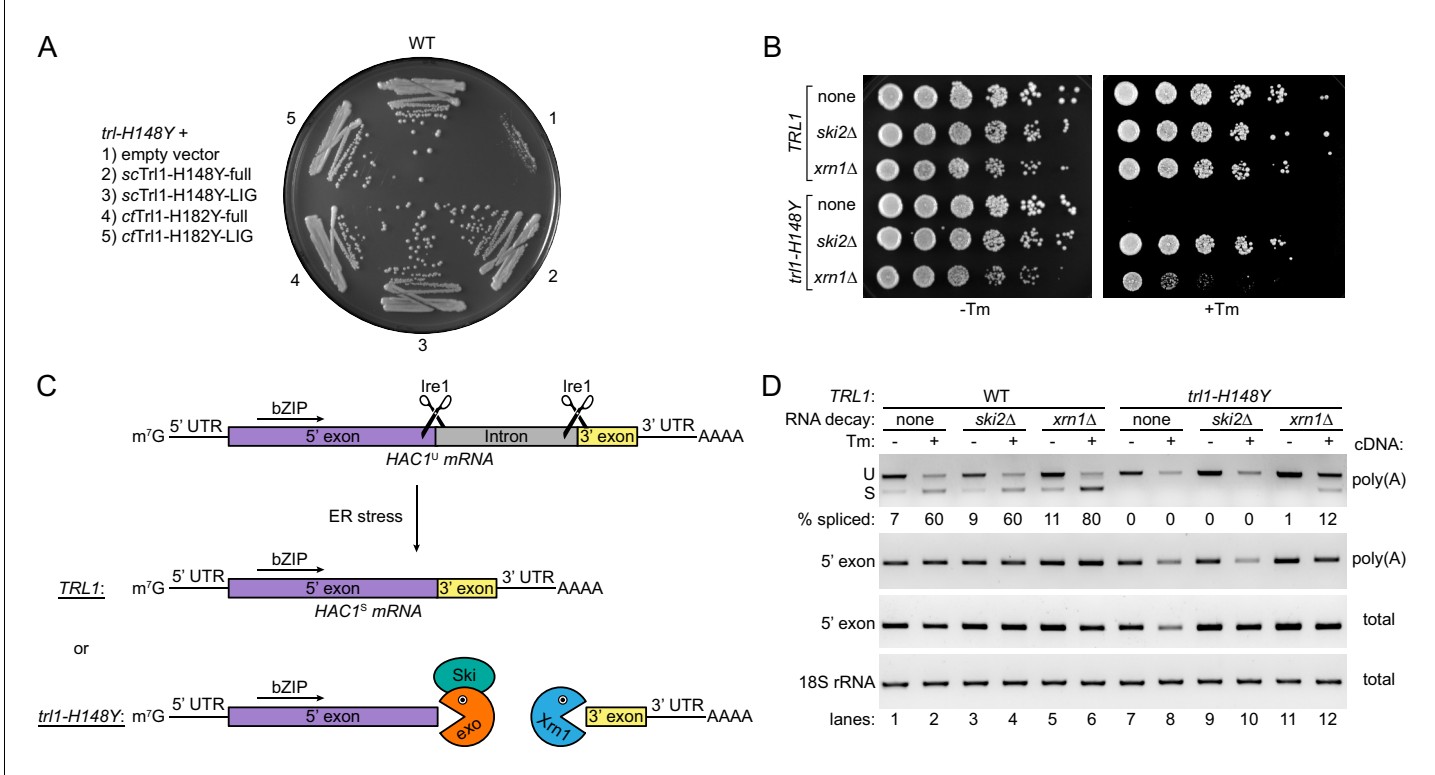

**Figure 5.** *HAC1* mRNA splicing competes with general RNA decay pathways. (**A**) Overexpression of various Trl1 constructs rescued the growth phenotype of *trl1-H148Y*. The constructs from *S. cerevisiae* harbored the H148Y mutation of the *trl1-H148Y* allele, whereas those from *C. thermophilum* harbored the equivalent H182Y mutation. Both the full-length proteins and the isolated ligase domains restored growth of the UPR mutant strain when streaked on tunicamycin-containing YPD plates. (**B**) Serial five-fold dilutions of wild-type and *trl1-H148Y* cells with wild-type or abrogated (*ski2Δ* or *xrn1Δ*) RNA decay pathways were spotted on YPD plates without (-Tm) or with (+Tm) 0.1 µg/ml tunicamycin and incubated at 30˚C for 2 days. (**C**) Model of *HAC1* mRNA splicing and the impact of RNA decay pathways on degradation of cleaved *HAC1* exons. Ire1 cleaves unspliced *HAC1* mRNA (*HAC1*$^U$) at the non-conventional splice sites upon ER stress. In wild-type yeast cells (*TRL1*), exon-exon ligation by Trl1 is the predominant reaction yielding spliced *HAC1* mRNA (*HAC1*$^S$). In the context of a kinetically compromised tRNA ligase (*trl1-H148Y*), the cleaved *HAC1* exons are degraded by exonucleolytic RNA decay pathways. The capped (m$^7$G) 5' exon is susceptible to degradation in 3'-to-5' direction by the RNA exosome (exo)/Ski complex (Ski); the 3' exon is degraded from its 5' end by Xrn1. (**D**) RT-PCR analysis of *HAC1* mRNA splicing in the same strains as in **B** without and with tunicamycin (Tm). The top panel shows priming outside the exon-intron junctions to distinguish unspliced (U) from spliced (S) *HAC1* mRNA. The relative amount of *HAC1*$^S$ (in %) is indicated below each lane. Both middle panels show the results for priming of the 5' exon. 18S rRNA was used as a control (bottom panel). The method of cDNA production is indicated on the right of each panel.

DOI: https://doi.org/10.7554/eLife.44199.010

The following figure supplement is available for figure 5:

**Figure supplement 1.** The rescue of UPR-deficiency in *trl1-H148Y* is dependent on *HAC1*.

DOI: https://doi.org/10.7554/eLife.44199.011

with previous results, upon ER stress we detected an overall reduction of unspliced *HAC1* mRNA and no *HAC1* mRNA splicing in *trl1-H148Y* cells (*Figure 5D*, lanes 7 and 8). Deletion of *XRN1* in *trl1-H148Y* cells restored splicing, albeit not to the same degree as in the WT strain (*Figure 5D*, lanes 11 and 12). We surmise that stabilization of the severed *HAC1* 3' exon in the absence of the major 5'-to-3' exonuclease Xrn1 allows enzymatically impaired Trl1-H148Y to catch up and ligate the *HAC1* exons to restore the UPR.

To our surprise, despite the complete phenotypic rescue shown above (*Figure 5B*), we did not detect spliced *HAC1* mRNA in *ski2Δ trl1-H148Y* cells upon ER stress (*Figure 5D*, lanes 9 and 10), but rather a reduction of unspliced *HAC1* mRNA, reminiscent of the parental *trl1-H148Y* mutant strain. We confirmed this result using RT-PCR to amplify the 5' exon only (*Figure 5D*, row 2, compare lanes 8 and 10). Both experiments did not account for RNA lacking a poly(A) tail since we used oligo(dT) primers to generate the cDNA templates. Interestingly, when we performed the same RT-PCR on

the 5' exon with cDNA templates from the entire cellular RNA (using random hexamers for priming), we now observed an increase in the amount of the *HAC1* 5' exon upon *SKI2* deletion, when compared to the parental *trl1-H148Y* strain (*Figure 5D*, row 3, compare lanes 8 and 10). Taken together, the RT-PCR analyses suggest that deletion of either *XRN1* or *SKI2* rescued the UPR-deficiency of the *trl1-H148Y* allele by different mechanisms: The genetic ablation of the major 5'-to-3' exonuclease Xrn1 restored *HAC1* mRNA splicing in the UPR-deficient yeast cells. By contrast, as we did not detect spliced *HAC1* mRNA in *ski2Δ trl1-H148Y* cells, abrogation of cytosolic RNA exosome by disruption of the Ski complex must have suppressed ER stress sensitivity solely via stabilization of the severed 5' exon.

## Rescue of stalled ribosomes initiates the UPR from the *HAC1* 5' exon fragment

Previous studies showed that a truncated form of Hac1 produced from an mRNA bearing a stop codon immediately after the 5' exon and hence lacking the entire C-terminal portion encoded by the 3' exon (Hac1_{trunc}) is a fundamentally still functional transcription factor that can induce the UPR (*Cox and Walter, 1996*; *Di Santo et al., 2016*). Since the stabilized, cleaved 5' exon in the *ski2Δ trl1-H148Y* strain reported here does not contain a stop codon, it should be subject to co-translational mRNA surveillance by no-go decay (NGD). This ribosome recycling pathway hinges on the splitting of ribosomal subunits by the non-canonical release factors Dom34 (Pelota in mammals) and Hbs1. We generated *DOM34* deletion strains to test for the possible involvement of NGD in translation of the severed *HAC1* 5' exon. Growth analysis of these strains showed that deletion of *DOM34*

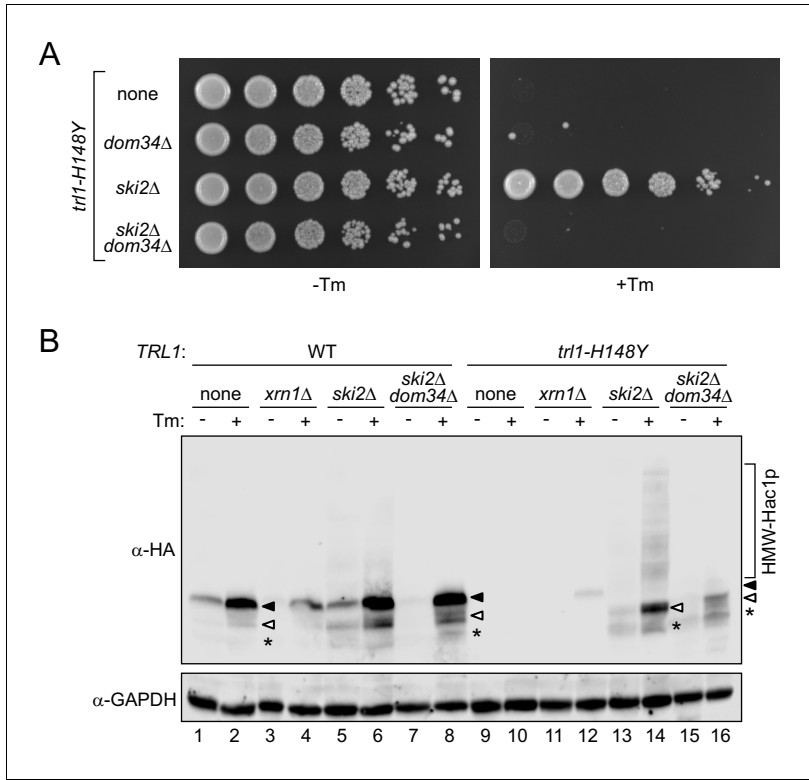

**Figure 6.** Ribosome rescue allows UPR signaling from the *HAC1* 5' exon fragment. (**A**) Evaluating the role of ribosome recycling on the *ski2Δ*-dependent growth rescue of *trl1-H148Y* cells under ER stress. Serial five-fold dilutions of *trl1-H148Y* cells with the indicated deletions of *SKI2* and/or *DOM34* were spotted on YPD plates without (-Tm) or with (+Tm) 0.1 μg/ml tunicamycin and incubated at 30˚C for 2 days. (**B**) Lysates of the indicated strains were prepared and immunoblotted for 3xHA-Hac1p and GAPDH as loading control. Full-length Hac1p is indicated by a closed arrowhead, truncated Hac1p by an open arrowhead. An even shorter minor third band is marked with an asterisk. The high molecular weight smear/banding pattern is indicated as HMW-Hac1p.
DOI: https://doi.org/10.7554/eLife.44199.012

re-sensitized *ski2Δ trl1-H148Y* cells to ER stress (*Figure 6A*). Thus, rescue of *trl1-H148Y* by depletion of Ski2 (i.e. cytosolic RNA exosome activity) is dependent on successful ribosome rescue at the 5' non-conventional splice site of the cleaved *HAC1* mRNA.

To further corroborate the findings from the growth and *HAC1* mRNA RT-PCR assays, we analyzed Hac1 levels by immunoblotting of extracts from cells expressing epitope tagged 3x-HA-Hac1. We confirmed the production of Hac1 upon ER stress in WT cells and the lack of its expression in *trl1-H148Y* cells (*Figure 6B*, lanes 1–2 and 9–10). Two distinct mechanisms prevent the accumulation of translation product from unspliced *HAC1* mRNA. A long-range base pairing interaction between the $HAC1^U$ intron and its 5' untranslated region (UTR) represses Hac1 protein production (*Rüegsegger et al., 2001*) by inhibiting translation initiation (*Sathe et al., 2015*). In addition, any translation products from the unspliced *HAC1* mRNA are efficiently degraded due to an in-frame degron encoded in the intron (*Di Santo et al., 2016*). As expected from the results shown above, upon ER stress induction we detected full-length Hac1 in *xrn1Δ trl1-H148Y* cells (*Figure 6B*, lane 12). The expression levels were lower than in WT cells, in accordance with the reduced amount of spliced *HAC1* mRNA produced in this strain (*Figure 5D*, compare lane 2 with lane 12). By contrast, while we detected no Hac1 in *ski2Δ trl1-H148Y* cells exposed to ER stress, we saw instead a faster migrating band (*Figure 6B*, lane 14), consistent with the expression of a C-terminally truncated form of Hac1 in these cells. In the same strain, we also detected a series of higher molecular mass smeared out bands, presumably a banding pattern resulting from ubiquitination. Interestingly, deletion of *SKI2* in the wild-type strain also led to an accumulation of truncated Hac1 and higher molecular mass laddering, albeit to a lesser extent than in the respective *trl1-H148Y* strain. This result indicated that RNA decay and ribosome-associated mRNA surveillance, while their effects are exacerbated in the UPR mutant strain, play an important role in maintaining the fidelity of *HAC1* mRNA splicing in wild-type cells.

## Discussion

Since the discovery of Trl1 as the tRNA ligase in *S. cerevisiae* by Abelson and co-workers, its atomic structure has remained elusive. Trl1 was later identified in a yeast genetic screen as an indispensable component of the UPR. The deficiency of the underlying H148Y mutation within Trl1-LIG to splice *HAC1* mRNA in response to ER stress posed an intriguing mystery. The lack of high-resolution structural data rendered the quest for mechanistic understanding of this phenotype challenging and thus the problem remained unsolved for more than two decades. In the present study, we present the crystal structure of Trl1-LIG and provide new insights into the intricate kinetic competition between *HAC1* mRNA splicing and mRNA decay.

The crystal structure of Trl1-LIG from *C. thermophilum* revealed a two-domain architecture comprised of a canonical adenylyltransferase domain and a C-terminal domain with a unique all-helical fold within the same enzyme superfamily. Previous structural and enzymatic data on T4Rnl1 showed that its C-terminal domain is not required to catalyze RNA ligation but confers specificity for tRNA repair (*El Omari et al., 2006*). Based on these observations and the structural similarities between the adenylyltransferase domains of *ct*Trl1-LIG and T4Rnl1, we suspect a similar function, that is conferring substrate specificity, of the C-terminal domain of *ct*Trl1-LIG. It will be interesting to see if the requirement for a 2' phosphate on the 5' exon is mediated by this unique domain.

A major motivation for our efforts towards an atomic structure of Trl1-LIG was to understand the separation of function resulting from the UPR mutant allele *trl1-H148Y*. The reduced ligation kinetics of Trl1-H148Y tip the balance almost completely in favor of degrading cleaved *HAC1* mRNA. Both major exonucleolytic decay pathways of the cytosol, Xrn1 and the RNA exosome with the Ski complex, are part of this kinetic competition of RNA processing enzymes, albeit with different roles.

Our data suggest that Xrn1 degrades the 3' exon fragment of cleaved *HAC1* and plays an important role in allowing efficient translation from the spliced mRNA. The *HAC1* intron in *S. cerevisiae* blocks translation initiation by forming complementary base pairing with the 5' UTR of the mRNA (*Rüegsegger et al., 2001*; *Sathe et al., 2015*). Degradation of the intron after splicing was reported as required for lifting the translational block and the production of Hac1 (*Mori et al., 2010*). Deletion of *XRN1*, even in the presence of WT Trl1, resulted in lower levels of Hac1 upon ER stress. Interestingly, Xrn1 also degrades tRNA introns during tRNA splicing, a process with equivalent biochemistry as *HAC1* mRNA splicing and shared functionality of Trl1 (*Wu and Hopper, 2014*). It

should be noted that in all cases phosphorylation of the 5' end, as catalyzed by the central polynucleotide kinase part of Trl1, is required to render the RNA molecules substrates for Xrn1. In consequence, this kinase-mediated decay poses a permanent competition for Trl1-mediated ligation of any RNA substrate in the cell. Degradation of tRNA introns in yeast has been previously reported to be catalyzed by Trl1-KIN and Xrn1 in an analogous two-step process (*Wu and Hopper, 2014*).

We showed that abrogation of the RNA exosome by deletion of *SKI2* rescued ER stress sensitivity of *trl1-H148Y*. However, splicing was not restored in *ski2Δ*; instead, stabilization of the isolated 5' exon fragment of *HAC1* allowed UPR signaling. Previous studies have shown by adding a stop codon after the 5' exon of *HAC1* that the 18 amino acids encoded in the 3' exon are dispensable for the transactivation function of Hac1p. In *trl1-H148Y ski2Δ*, the truncated form of Hac1p would need to be translated and released from an mRNA that lacks a stop codon. Indeed, we could demonstrate, that Hac1$_{trunc}$ is produced from the isolated *HAC1* 5' exon in sufficient amounts to initiate the transcriptional response. The nascent chain release in this context is dependent on ribosome splitting by the NGD pathway. This finding is in accordance with a previous study that identified *HAC1* mRNA as the strongest Dom34 target by ribosome profiling (*Guydosh and Green, 2014*). A recent study demonstrated that RIDD in *Schizosaccharomyces pombe* also depends on NGD-mediated ribosome rescue to allow degradation of cleaved mRNAs by the Ski complex/exosome (*Guydosh et al., 2017*; *Kimmig et al., 2012*). Together, these findings establish a crucial role of NGD in both functional outputs of the Ire1 branch of the UPR. Interestingly, production of Hac1$_{trunc}$ coincides with a high-molecular weight laddering that we interpret as ubiquitination of the truncated form. We surmise that the *HAC1* mRNA is an endogenous substrate of the recently characterized ribosome quality control (RQC) complex (*Brandman et al., 2012*; *Shen et al., 2015*).

The UPR is of crucial importance for cellular homeostasis. In yeast, Hac1 initiates a major transcriptional response that re-shapes the entire ER and upregulates hundreds of target genes comprising ~5% of the genome (*Travers et al., 2000*). Thus, tight regulation of the response is of great importance for cell physiology. Our data show, that deletion of *SKI2* leads to the accumulation of Hac1$_{trunc}$ – even in the context of WT Trl1. It is very likely that single cleavage by Ire1 occurs frequently and that RNA decay pathways assist in degrading these *HAC1* mRNA fragments to prevent premature UPR signaling. We conclude that tight regulation of the UPR in yeast is achieved by the interplay of several control mechanisms to which this study adds RNA decay and surveillance by mRNA quality control factors.

Trl1 homologs are present in all human fungal pathogens. They are essential enzymes for them due to their role in tRNA splicing. They also represent promising targets for antifungal drug discovery because their structure and mechanism are distinct from the RTCB-type tRNA ligases in metazoans (as well as archaea and many bacteria). The absence of human homologs of Trl1-LIG could therefore be explored as a promising new drug target in antifungal therapy, whose development has largely stalled in recent years. Together with the recently published structure of *Candida albicans* Trl1 kinase domain Trl1-KIN (*Remus et al., 2017*), our study presents new opportunities for inhibitor screens guided by atomic-resolution Trl1 structures.

## Materials and methods

**Key resources table**

| Reagent type (species) or resource | Designation | Source or reference | Identifiers | Additional information |
|---|---|---|---|---|
| Gene (*Chaetomium thermophilum*) | TRL1 | *Chaetomium thermophilum* genome resource | Ct:CTHT_0034810 | |
| Gene (*Saccharomyces cerevisiae*) | TRL1 | *Saccharomyces* Genome Database | SGD:S000003623 | |
| Gene (*Saccharomyces cerevisiae*) | IRE1 | *Saccharomyces* Genome Database | SGD:S000001121 | |

*Continued on next page*

*Continued*

| Reagent type (species) or resource | Designation | Source or reference | Identifiers | Additional information |
|---|---|---|---|---|
| Gene (*Saccharomyces cerevisiae*) | *HAC1* | *Saccharomyces* Genome Database | SGD:S000001863 | |
| Antibody | anti-HA (mouse monoclonal) | Sigma | Sigma:H3663; RRID:AB_262051 | (1:2000) |
| Antibody | anti-GAPDH (rabbit polyclonal) | abcam | abcam:ab9485; RRID:AB_307275 | (1:1000) |
| Antibody | goat anti-mouse (goat polyclonal, HRP conjugate) | Promega | Promega:W4021; RRID:AB_430834 | (1:10000) |
| Antibody | goat anti-rabbit (rabbit polyclonal, HRP conjugate) | Promega | Promega:W4011; RRID:AB_430833 | (1:10000) |
| Recombinant DNA reagent | HAC1$^U$-508-pBSK- | PMID:9323131 | lab archive:pPW0386; paper:pCF187 | *HAC1* in vitro transcript (508 nucleotides) |
| Sequence-based reagent | Cy5-5' exon *HAC1* RNA oligonucleotide | IDT | | 5'-Cy5-CGUAA UCCAG-3'-PO4 |
| Sequence-based reagent | Cy5-3' exon *HAC1* RNA oligonucleotide | IDT | | 5'-PO4-AAGCG CAGUC-Cy5 |
| Sequence-based reagent | FAM-5' exon *HAC1* RNA oligonucleotide | IDT | | 5'-FAM-CGUAAU CCAG-3'-PO4 |
| Sequence-based reagent | 3' exon *HAC1* RNA oligonucleotide | IDT | | 5'-PO4-AAGCGCAGUC AGGUUUGAAU-3' |
| Sequence-based reagent | oligo a for RNase H assay | IDT | | TAATCACGGC GGACAGTA |
| Sequence-based reagent | oligo b for RNase H assay | IDT | | TTGAAGGTAC TTTAACCG |
| Sequence-based reagent | oligo-(dT)$_{12-18}$ | Thermo Fisher Scientific | Thermo Fisher Scientific:18418012 | |
| Sequence-based reagent | random hexamers | Thermo Fisher Scientific | Thermo Fisher Scientific:N8080127 | |
| Peptide, recombinant protein | *ct*Trl1 [full length] | This paper | | purified from *E. coli* BL21-RIL cells |
| Peptide, recombinant protein | *ct*Trl1-LIG | This paper | | purified from *E. coli* BL21-RIL cells |
| Peptide, recombinant protein | *sc*Trl1-LIG-WT | This paper | | purified from *E. coli* BL21-RIL cells |
| Peptide, recombinant protein | *sc*Trl1-LIG-H148Y | This paper | | purified from *E. coli* BL21-RIL cells |
| Peptide, recombinant protein | *sc*Trl1-CPD | This paper | | purified from *E. coli* BL21-RIL cells |
| Peptide, recombinant protein | *sc*Ire1-KR32 | This paper | | purified from *E. coli* BL21-RIL cells |
| Peptide, recombinant protein | RtcA | This paper | | purified from *E. coli* BL21-RIL cells |
| Peptide, recombinant protein | His$_{10}$-HRV-3C protease | This paper | | purified from *E. coli* BL21-RIL cells |
| Peptide, recombinant protein | T4 polynucleotide kinase (3' phosphatase minus) | New England BioLabs | New England BioLabs:M0236S | |
| Peptide, recombinant protein | Proteinase K | Thermo Fisher Scientific | Thermo Fisher Scientific:25530049 | |

*Continued*

| Reagent type (species) or resource | Designation | Source or reference | Identifiers | Additional information |
|---|---|---|---|---|
| Peptide, recombinant protein | RNAse H | New England BioLabs | New England BioLabs:M0297S | |
| Peptide, recombinant protein | SuperScript II reverse transcriptase | Thermo Fisher Scientific | Thermo Fisher Scientific:18064014 | |
| Chemical compound, drug | AMPcPP | Sigma | Sigma:M6517 | |
| Chemical compound, drug | 4μ8C | Matrix Scientific | Matrix Scientific: 037985; CAS:14003-96-4 | |
| Chemical compound, drug | Tunicamycin | Sigma | Sigma:T7765 | |
| Chemical compound, drug | SYBR Gold nucleic acid stain | Thermo Fisher Scientific | Thermo Fisher Scientific:S11494 | |

## Cloning of expression constructs

Full-length *ct*Trl1 (NCBI Entrez Gene ID: 18257519, CTHT_0034810 tRNA ligase-like protein) was cloned from *C. thermophilum* cDNA into pET15b (EMD Millipore; pPW3206) as described previously (*Peschek et al., 2015*). The expression construct for *ct*Trl1-LIG (residues 1–414; pPW3207) was generated by PCR-based introduction of a stop codon (oligo: 5'-GCTAGCGTGTAGCGTGACATCATTC-3') into the pET15b vector using site-directed mutagenesis. The expression construct for *ct*Trl1-CPD (residues 576–846; pPW3208) was generated by cloning the target sequence into pET15b using PCR amplification, followed by restriction digestion and DNA ligation. Similarly, the expression constructs for WT (pPW3209) and H148Y (pPW3210) *sc*Trl1-LIG (residues 1–388) were cloned into pET47b using the restriction enzymes SmaI and BamHI. The expression construct for *E. coli* RtcA (pPW3420) was generated using the In-Fusion HD Cloning Plus kit (Takara Bio). For the In-Fusion reaction, pET28a (EMD Millipore) was linearized by NdeI (New England BioLabs) and the RtcA insert was amplified from *E. coli* DNA by PCR. All described constructs were confirmed by DNA sequencing. See *Supplementary file 1* for a complete list of plasmids used in this study.

## Protein expression and purification

All recombinant proteins in this study were expressed in *E. coli* BL21-CodonPlus (DE3)-RIL (Agilent Technologies). The cells were grown in Luria broth medium including the appropriate antibiotic at 37°C, expression was induced by 1 mM isopropyl-1-thio-β-D-galactopyranoside (IPTG), cells were harvested by centrifugation and lysed using an EmulsiFlex-C3 (Avestin) high-pressure homogenizer (exceptions to the above as noted).

Recombinant, His$_6$-tagged *ct*Trl1 was purified by Ni$^{2+}$ affinity chromatography using a HisTrap FF column (GE Healthcare Life Sciences), followed by size-exclusion chromatography using a HiLoad 16/60 Superdex200 pg column (GE Healthcare Life Sciences) in 20 mM Tris/HCl pH 7.1, 300 mM NaCl, and 1 mM MgCl$_2$ (*Peschek et al., 2015*).

Selenomethionine (SeMet)-substituted His$_6$-tagged *ct*Trl1-LIG (residues 1–414) was expressed using feedback inhibition of methionine biosynthesis by adding selected amino acids to the *E. coli* culture prior to induction. In detail, cells were grown in standard M9 minimal medium supplemented with thiamine (0.5% w/v) and trace elements. When the OD$_{600}$ reached 0.6, the temperature was lowered to 28°C and the feedback-inhibition amino acids mix (1 g/l of lysine, threonine and phenylalanine, 0.5 g/l leucine, isoleucine and valine, and 0.5 g/l L(+)-SeMet) was added and, after 30 min, the cells were induced with IPTG. After 16 hr expression, cells were harvested and disrupted in lysis buffer (40 mM NaH$_2$PO$_4$/Na$_2$HPO$_4$ pH 7.4, 1.2 M NaCl, 25 mM imidazole, 5 mM dithiothreitol [DTT]) plus protease inhibitor (cOmplete EDTA-free protease inhibitor cocktail, Roche). The cleared supernatant was applied to a Ni$^{2+}$ affinity chromatography column (HisTrap FF 5 ml, GE Healthcare Life Sciences) and *ct*Trl1-LIG eluted with a 25–500 mM imidazole gradient (same as lysis buffer). LIG-containing fractions were dialysed against AEX buffer (20 mM HEPES/NaOH pH 8, 10 mM NaCl, 5 mM DTT) and further purified by anion exchange chromatography (HiTrap Q HP 5 ml, GE Healthcare Life

Sciences) using a 10–1000 mM NaCl gradient in AEX buffer. Finally, the protein was applied to size-exclusion chromatography using a HiLoad 16/60 Superdex200 pg column (GE Healthcare Life Sciences) in SEC buffer (10 mM HEPES/NaOH pH 7.5, 100 mM NaCl, 5 mM DTT), concentrated and flash frozen in $N_2$ (l).

His$_6$-tagged $ct$Trl1-CPD (residues 576–846) was expressed at 30°C for 5 hr. Cells were harvested and disrupted in lysis buffer (50 mM Tris/HCl pH 7.5, 1 M NaCl, 20 mM imidazole, 2 mM DTT) plus protease inhibitor (cOmplete EDTA-free protease inhibitor cocktail, Roche). The cleared supernatant was applied to a $Ni^{2+}$ affinity chromatography column (HisTrap FF 5 ml, GE Healthcare Life Sciences) and $ct$Trl1-CPD eluted with a 20–500 mM imidazole gradient (same as lysis buffer). CPD-containing fractions were further purified by size-exclusion chromatography using a HiLoad 16/60 Superdex200 pg column (GE Healthcare Life Sciences) in SEC buffer (10 mM Tris/HCl pH 7.5, 100 mM NaCl, 20 mM imidazole, 1 mM tris(2-carboxyethyl)phosphine hydrochloride [TCEP]), concentrated and flash frozen in $N_2$ (l).

His$_6$-tagged $sc$Trl1-LIG-WT and -H148Y (residues 1–388) were expressed at 20°C for 16 hr. Cells were harvested and disrupted in lysis buffer (40 mM HEPES/NaOH pH 7.5, 1 M NaCl, 25 mM imidazole, 2 mM DTT, 5% glycerol) plus protease inhibitor (cOmplete EDTA-free protease inhibitor cocktail, Roche). The cleared supernatant was applied to a $Ni^{2+}$ affinity chromatography column (HisTrap FF 5 ml, GE Healthcare Life Sciences) and $sc$Trl1-LIG eluted with a 25–500 mM imidazole gradient (same as lysis buffer). LIG-containing fractions were dialysed against 3C cleavage buffer (25 mM HEPES/NaOH pH 8, 75 mM NaCl, 2 mM DTT, 5% glycerol) followed by proteolytic removal of the His$_6$-tag by incubation with recombinant His$_{10}$-tagged human rhinovirus 3C (HRV3C) protease. To separate the cleaved target protein from the uncleaved fraction and HRV3C protease, another round of $Ni^{2+}$ affinity chromatography was performed. The flow-through was dialysed against AEX buffer (25 mM HEPES/NaOH pH 8, 25 mM NaCl, 2 mM DTT, 5% glycerol) and further purified by anion exchange chromatography (HiTrap Q HP 5 ml, GE Healthcare Life Sciences) using a 25–1000 mM NaCl gradient in AEX buffer. Finally, the protein was applied to size-exclusion chromatography using a HiLoad 16/60 Superdex200 pg column (GE Healthcare Life Sciences) in SEC buffer (25 mM HEPES/NaOH pH 7.5, 150 mM NaCl, 1 mM TCEP, 5% glycerol), concentrated and flash frozen in $N_2$ (l).

The cytosolic kinase/RNase portion of yeast Ire1 including 32 residues of the linker region (Ire1-KR32) was recombinantly expressed in $E.\ coli$ as glutathione S-transferase (GST) fusion protein and purified as described previously (*Korennykh et al., 2009*).

His$_6$-tagged $E.\ coli$ RtcA was expressed at 37°C for 4 hr. Cells were harvested and disrupted in lysis buffer (50 mM Tris/HCl pH 7.5, 300 mM NaCl, 25 mM imidazole, 2 mM DTT, 10% glycerol) plus protease inhibitor (cOmplete EDTA-free protease inhibitor cocktail, Roche). The cleared supernatant was applied to a $Ni^{2+}$ affinity chromatography column (HisTrap FF 5 ml, GE Healthcare Life Sciences) and RtcA eluted with a 20–500 mM imidazole gradient (same as lysis buffer). Protein-containing fractions were diluted to 50 mM NaCl and further purified by anion exchange chromatography (HiTrap Q HP 5 ml, GE Healthcare Life Sciences) using a 50–1000 mM NaCl gradient. Finally, the protein was applied to size-exclusion chromatography using a HiLoad 16/60 Superdex75 pg column (GE Healthcare Life Sciences) in SEC buffer (20 mM Tris/HCl pH 7.5, 100 mM NaCl, 1 mM DTT, 5% glycerol), concentrated and flash frozen in $N_2$ (l).

## In vitro splicing assays

For endonucleolytic cleavage, in vitro transcribed, polyacrylamide gel electrophoresis (PAGE)-purified, refolded $HAC1^U$-508 RNA (*Gonzalez et al., 1999*) was incubated at 20 ng/µl with Ire1KR32 at 1 µM in cleavage buffer (20 mM HEPES/NaOH pH 7.5, 70 mM NaCl, 2 mM Mg(OAc)$_2$, 1 mM TCEP, and 5% glycerol). For splicing, ligation of the Ire1-cleaved $HAC1^U$-508 RNA was initiated with 500 nM $ct$Trl1 (1 mM ATP, 1 mM GTP) or the indicated amounts of $sc$Trl1-LIG. To allow ligation by Trl1-LIG, the ends of the cleaved RNA were first modified by 1 µM $ct$Trl1-CPD and 667 units/ml T4 polynucleotide kinase (3' phosphatase minus, New England BioLabs) plus 4 mM ATP. Stop solution (10 M urea, 0.1% SDS, 1 mM EDTA) was added at five-fold excess to stop the reactions. Samples were then denatured by heating at 80°C for 3 min and analyzed by 6% TBE–urea gels (Thermo Fisher Scientific) stained with SYBR Gold nucleic acid stain (Thermo Fisher Scientific).

## RNase H assay

Oligonucleotide-guided RNase H digestion was used to identify RNA products of *HAC1* in vitro splicing. The assay was performed similarly as described previously (*Mori et al., 2010*). After completion of the ligation reaction, 250 pmol of antisense DNA oligonucleotides were added to a total volume of 7.5 µL. The oligonucleotides were designed to hybridize with the *HAC1* intron (oligo a: TAATCACGGCGGACAGTA; oligo b: TTGAAGGTACTTTAACCG). Samples were heated to 75°C, incubated at 43°C for 10 min to allow hybridization, then slowly cooled to 37°C and incubated with 5 U (addition of 1 µL) of RNase H (5,000 U/ml; New England BioLabs) for 30 min. After proteinase K treatment, the samples were analyzed by PAGE as described in the previous section.

## Fluorescence-based RNA ligation and binding assays

All *HAC1*-derived RNA oligonucleotides were purchased from Integrated DNA Technologies. The 5′ exon and 3′ exon RNA substrates were synthesized with 3′ and 5′ phosphate groups at their respective splice site ends. RNA ligation kinetics were measured using RNA oligonucleotides based on the exon sequences closest to the splice sites. We used a 10 nt long 5′ exon fragment with a terminal fluorescein (FAM) fluorophore (5′-FAM-CGUAAUCCAG-3′-$PO_4$) and a 20 nt long 3′ exon fragment (5′-$PO_4$-AAGCGCAGUCAGGUUUGAAU-3′). The 5′exon fragment was modified by RtcA and *ct*Trl1-CPD in the presence of ATP to yield 2′ phosphate/3′ hydroxyl ends. Both RNA substrates were PAGE-purified prior to the assay. 250 nM *sc*Trl1-LIG (WT or H148Y) were incubated with 500 nM of 3′ exon oligonucleotide and 0.5 mM ATP in assay buffer (20 mM Tris/HCl pH 7.5, 150 mM NaCl, 10 mM $MgCl_2$, 1 mM TCEP, 5% glycerol) for 10 min at 30°C. The ligation kinetics were initiated by addition of 50 nM FAM-labeled 5′ exon fragment. Aliquots were taken at the indicated timepoints and mixed with a 3-fold excess of Stop solution. The ligation reaction was analyzed by 15% TBE–urea gels (Thermo Fisher Scientific) and imaged for FAM fluorescence using a Typhoon 9400 Variable Mode Imager (GE Lifesciences). The relative intensity of the bands was determined by densitometry using ImageJ (*Schindelin et al., 2012*; *Schneider et al., 2012*). Kinetic data were fitted to a first-order model using Origin (Version 2019, OriginLab Corporation).

Affinity measurements were performed using RNA oligonucleotides based on the *HAC1* exons' most proximal 10 nucleotides to the splice site. The 5′ exon and 3′ exon RNA substrates were modified with Cy5 fluorophores at their distal termini (5′ exon RNA: 5′-Cy5-CGUAAUCCAG-3′-$PO_4$; 3′ exon RNA: 5′-$PO_4$-AAGCGCAGUC-Cy5). The 5′exon fragment was modified by RtcA and *ct*Trl1-CPD in the presence of ATP to yield 2′ phosphate/3′ hydroxyl ends. Both RNA substrates were PAGE-purified prior to the binding assay. Binding was measured by changes in fluorescence using a Monolith NT.115 (NanoTemper). The RNA substrate concentration was held constant at 10 nM and the concentration of *sc*Trl1-LIG. RNA binding data were fitted to a Hill model using Origin (Version 2019, OriginLab Corporation).

## Crystallization, data collection and structure determination

Crystals of SeMet-*ct*Trl1-LIG were grown at 20°C by hanging drop vapor diffusion. Two volumes of SeMet-ctTrl1-LIG (12.5 mg/ml), 2 mM AMPcPP (Sigma, M6517) and 2 mM $MgCl_2$ were mixed with one volume of reservoir solution containing 1.7 M ammonium sulfate, 200 mM NaCl, 100 mM HEPES (pH 7.3) and 5 mM DTT. Obtained crystals were harvested directly from the crystallization drops, cryoprotected by soaking in reservoir solution including 25% glycerol and frozen in liquid nitrogen. X-ray diffraction data were collected at the Advanced Photon Source beamline 23ID-D and processed using MOSFLM (*Battye et al., 2011*) and AIMLESS (*Evans and Murshudov, 2013*) as implemented in the CCP4 software suite (*Winn et al., 2011*).

Phasing and initial model building were carried out using CRANK2 for automated structure solution (*Skubák and Pannu, 2013*). The phases were obtained using single-wavelength anomalous dispersion (SAD) data from a single crystal with SHELX (*Sheldrick, 2010*). The initial model was built using BUCCANEER (*Cowtan, 2006*) and used to solve the structure from a second high-resolution dataset. Interactive model building was performed using COOT (*Emsley et al., 2010*) and the structure was refined with REFMAC5 (*Murshudov et al., 2011*) including TLS refinement. The model optimization and quality assessment was carried out with PDB-REDO (*Joosten et al., 2014*) and MOLPROBITY (*Davis et al., 2004*). All structure figures were generated using PyMOL (The PyMOL Molecular Graphics System, Version 2.0 Schrödinger, LLC) and electrostatic surface potential

calculations were performed using PDB2PQR (*Dolinsky et al., 2004*) and APBS (*Baker et al., 2001*). Ligand interactions were analyzed using the protein-ligand interaction profiler (PLIP) web service (*Salentin et al., 2015*). A composite omit map was generated in Phenix (*Adams et al., 2010*) to exclude model bias and to verify ligand density.

## Yeast strains and growth conditions

All yeast strains in this study were based on *S. cerevisiae* W303 that was made *TRP1* and *ADE2* by repairing the endogenous auxotrophy. All strains are listed in *Supplementary file 2*. Deletion of non-essential genes was performed in haploid cells, deletion of *TRL1* was performed in diploid cells. Each deletion mutant was generated using a PCR-based homologous recombination method as described previously (*Janke et al., 2004*). Yeast transformations were performed by the standard LiOAc/single stranded carrier DNA/PEG method, cells were recovered on YPD medium and grown on the appropriate selection medium. Functional complementation of *trl1Δ* and *trl1-H148Y* strains was performed by transforming the strain of interest with different Trl1 constructs cloned into the p416ADH (ATCC 87376; pPW3211) yeast expression vector (pPW3212-3215; see *Supplementary file 1* for a complete list of plasmids).

For growth tests under ER stress, 5-fold serial dilutions were spotted onto either standard YPD plates (-Tm) or YPD plates containing the indicated concentration of tunicamycin (Sigma, T7765) dissolved in DMSO (+Tm). All growth tests were performed as three biological replicates; representative images are shown in the respective figures.

For immunoblot analysis, endogenous *HAC1* was N-terminally tagged with a 3x hemagglutinin (HA) tag (n-YPYDVPDYAGYPYDVPDYAGSYPYDVPDYA-c) in the strains of interest using the *Delitto perfetto* method (*Storici et al., 2001*).

## RNA extraction and RT-PCR

Total RNA was prepared from yeast using phenol-chloroform extraction followed by ethanol precipitation. The extracted RNA was resolubilized in water, the quality was assessed by agarose gel electrophoresis and the concentration determined by absorbance spectroscopy. For RT-PCR, cDNA was generated using SuperScript II reverse transcriptase (Thermo Fisher Scientific) with oligo-$(dT)_{12-18}$ or random hexamers primers (both Thermo Fisher Scientific) from 1 µg of total RNA. The cDNAs of interest were amplified from the reverse transcription reactions (1:100 dilution) with Phusion polymerase (Thermo Scientific) using the appropriate primer sets and optimal cycle numbers (see *Supplementary file 3* for primer sequences and cycle numbers). RT-PCR reactions were analyzed by agarose gel electrophoresis.

## Yeast protein extract preparation and immunoblotting

Extraction of yeast proteins was performed as described elsewhere (*Kushnirov, 2000*). Proteins were separated by SDS-PAGE (any kD Criterion TGX stain-free gels, Bio-Rad) and transferred onto Protran 0.2 µm pore nitrocellulose membrane (Perkin Elmar). Blots were probed with the following antibodies diluted in TBS-T buffer containing 5% nonfat dry milk: mouse anti-HA (1:2,000; Sigma H3663), rabbit anti GAPDH (1:1,000; abcam ab9485), HRP conjugated goat anti-mouse IgG (1:10,000; Promega W4021), HRP conjugated goat anti-rabbit IgG (1:10,000; Promega W4011). Blots were developed using SuperSignal West Dura Extended Duration Substrate (Pierce, Life Technologies) and chemiluminescence was detected on a ChemiDoc XRS + imager (Bio-Rad).

## Data availability

Structural coordinates for the *ct*Trl1-LIG structure have been deposited in the Protein Data Bank under accession code 6N67.

## Acknowledgements

We thank Christof Osman, Voytek Okreglak, Weihan Li, and members of the Walter Lab for their valuable help and insightful discussion; Damian Ekiert, Oren Rosenberg and Lan Wang for helpful advice regarding X-ray crystallography. We would like to thank the 2016 CCP4 School at Argonne including all its instructors for help and guidance to JP during data collection, initial data analysis

and structure determination. We thank the entire staff at GM/CA@APS, which has been funded in whole or in part with Federal funds from the National Cancer Institute (ACB-12002) and the National Institute of General Medical Sciences (AGM-12006). This research used resources of the Advanced Photon Source, a US Department of Energy (DOE) Office of Science User Facility operated for the DOE Office of Science by Argonne National Laboratory under Contract No. DE-AC02-06CH11357. JP acknowledges a long-term fellowship from the Human Frontier Science Program. This work was supported by NIH grant GM032384 to PW. PW is an Investigator of the Howard Hughes Medical Institute.

## Additional information

### Funding

| Funder | Grant reference number | Author |
| --- | --- | --- |
| Human Frontier Science Program | Postdoctoral Fellowship | Jirka Peschek |
| Howard Hughes Medical Institute | HHMI826735-0012 | Peter Walter |
| National Institute of General Medical Sciences | R01, GM032384 | Peter Walter |

The funders had no role in study design, data collection and interpretation, or the decision to submit the work for publication.

### Author contributions

Jirka Peschek, Conceptualization, Data curation, Formal analysis, Investigation, Visualization, Methodology, Writing—original draft, Project administration, Writing—review and editing; Peter Walter, Conceptualization, Resources, Data curation, Supervision, Funding acquisition, Writing—review and editing

### Author ORCIDs

Jirka Peschek (iD) https://orcid.org/0000-0001-8158-9301
Peter Walter (iD) http://orcid.org/0000-0002-6849-708X

### Decision letter and Author response

Decision letter https://doi.org/10.7554/eLife.44199.022
Author response https://doi.org/10.7554/eLife.44199.023

## Additional files

### Supplementary files

• Supplementary file 1. List of plasmids.
DOI: https://doi.org/10.7554/eLife.44199.013

• Supplementary file 2. Yeast strains (generated in this study).
DOI: https://doi.org/10.7554/eLife.44199.014

• Supplementary file 3. RT-PCR primers and experimental settings.
DOI: https://doi.org/10.7554/eLife.44199.015

• Transparent reporting form
DOI: https://doi.org/10.7554/eLife.44199.016

### Data availability

Structural coordinates for the ctTrl1-LIG structure have been deposited in the Protein Data Bank under accession code 6N67.

The following dataset was generated:

| Author(s) | Year | Dataset title | Dataset URL | Database and Identifier |
|---|---|---|---|---|
| Peschek J, Walter P | 2019 | Structural coordinates for the ctTrl1-LIG structure | http://www.rcsb.org/structure/6N67 | Protein Data Bank, 6N67 |

The following previously published dataset was used:

| Author(s) | Year | Dataset title | Dataset URL | Database and Identifier |
|---|---|---|---|---|
| Amlacher S, Sarges P, Flemming D, van Noort V, Kunze R, Devos DP, Arumugam M, Bork P, Hurt E | 2011 | Chaetomium thermophilum var. thermophilum DSM 1495 | https://www.ncbi.nlm.nih.gov/bioproject/47065 | NCBI BioProject, PRJNA47065 |

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
