## [Decision Letter]

Thank you for submitting your article "tRNA ligase structure reveals kinetic competition between non-conventional mRNA splicing and mRNA decay" for consideration by *eLife*. Your article has been reviewed by two peer reviewers, and the evaluation has been overseen by Nahum Sonenberg, the Reviewing Editor and Cynthia Wolberger as the Senior Editor. The following individual involved in review of your submission has agreed to reveal their identity: Javier Martinez (Reviewer #2).

The reviewers have discussed the reviews with one another and the Reviewing Editor has drafted this decision to help you prepare a revised submission.

Summary:

The authors studied the mechanism by which *HAC1* pre-mRNA is spliced in the cytoplasm of yeast cells, which is triggered by UPR. To this end, the authors solved the crystal structure of the *Saccharomyces cerevisiae* Trl1 homolog from Chaetomium thermophilum. They exploited in particular the His-to-Tyr mutation at Trl1 position 148 (within the ligase domain), which abolishes UPR signaling but not splicing, gaining important insight into the *HAC1* pre-mRNA splicing mechanism by demonstrating kinetic competition between non-conventional mRNA splicing and RNA degradation, which effects the outcome of Ire1-initiated mRNA processing. Thus, the mRNA decay and surveillance mechanisms interact to ensure fidelity of non-conventional mRNA splicing during the UPR.

Essential revisions:

Both reviewers agree that the paper presents an interesting study, which would warrant publication in *eLife* if amended in response to the reviewers' critiques. The Reviewing Editor concurs with the reviewers' opinions. As you can see in the comments below both reviewers believe that the key results are shown in Figure 3, but they argue that some issues related to these results should be experimentally addressed.

1) What is the basis of the reduction in splicing efficiency in the trl1-H148Y enzyme? Does the enzyme have a lower affinity for specific RNA fragments? An answer to this question might provide a link to the understanding of the enhanced sensitivity to Xrn1 or the exosome. This experiment could be readily done.

2) What is the band marked with an asterisk in Figure 1D, lane 3?

3) The results in Figure 4 should be quantified.

4) Discussion: Where is it shown that Xrn1 degrades the 3' exon?

---

## [Author Response]

Essential revisions:Both reviewers agree that the paper presents an interesting study, which would warrant publication in eLife if amended in response to the reviewers' critiques. The Reviewing Editor concurs with the reviewer' opinions. As you can see in the comments below both reviewers believe that the key results are shown in Figure 3, but they argue that some issues related to these results should be experimentally addressed.1) What is the basis of the reduction in splicing efficiency in the trl1-H148Y enzyme? Does the enzyme have a lower affinity for specific RNA fragments? An answer to this question might provide a link to the understanding of the enhanced sensitivity to Xrn1 or the exosome. This experiment could be readily done.

We agree with the reviewers’ assessment that a detailed mechanistic understanding of the reduced splicing (ligation) efficiency of the H148Y mutant will add to our understanding of the Trl1 ligation mechanism. We tested several different in vitro approaches to determine binding affinities for both exons. We measured affinities by monitoring fluorescence changes of labeled RNA oligonucleotides upon binding to *sc*Trl1-LIG. Unexpectedly, these experiments yielded very similar affinities for WT and mutant Trl1-LIG towards 5’ and 3’ exon fragment (differing by only ~1.4 and ~1.1-fold, respectively), respectively). We included these fluorescence binding measurements as new Figure 4—figure supplement 1.

We conclude from this work that, if binding affinities are not significantly impeded, the reduced activity of the mutant enzyme must result from a changed catalytic activity. To confirm the reduced ligation kinetics (efficiency) of the Trl1-H148Y enzyme (Figure 4B), we performed time-resolved in vitro ligation experiments with labeled RNA oligonucleotides. These results confirmed the effect of the His-to-Tyr mutation, causing a ~2-fold reduced apparent rate constants. Interestingly, the mutant ligase is not only a 2x slower enzyme but also plateaued at a >5-fold lower amount of ligation product compared to the wild-type enzyme. These results are depicted as new Figures 4C (PAGE gel) and 4D (quantification and fitting). Taken together, these results strongly support our kinetic competition model.

2) What is the band marked with an asterisk in Figure 1D, lane 3?

The * demarks a circularized tandem *HAC1* intron. We now added experimental evidence that this is indeed the case. To this end, we used antisense DNA oligos complementary to either the presumptive ligation (circularization) site or the middle of the intron. After annealing to the RNA products of the in vitro splicing reaction and treatment with RNase H, which specifically hydrolyzes the phosphodiester bonds of RNA in RNA/DNA hybrids, the band in question disappeared and a band at the exact position of the severed intron (252 nt) re-appeared. We conclude that * represents two cleaved introns that were circularized by two intermolecular ligation events at their 5’ and 3’ ends. The results are depicted in Figure 2—figure supplement 1. We added the RNase H assay to the *Materials and methods*.

3) The results in Figure 4 should be quantified.

We quantified the *HAC1* splicing results and added the results to Figure 5 (previously Figure 4).

4) Discussion: Where is it shown that Xrn1 degrades the 3' exon?

We softened the statement by changing “Our data show…” to “Our data suggest…”. Based on our RT-PCR results (Figure 5D), we could show that levels of the 5’ exon are not affected by the presence or absence of Xrn1. Although we did not directly “show” the degradation of the 3’ exon by Xrn1, it is the only *HAC1* mRNA component left in the non-conventional splicing reaction. Our RT-PCR data demonstrate that deletion of *XRN1* restores splicing, which requires a fully intact 3’ exon as substrate.